

# Compatibility of endoscopic examination using i-scan technology with histopathology results in laryngeal carcinoma: prospective observational study

Gracia Cintia Massie, Agung Dinasti Permana, Shinta Fitri Boesoirie, Lina Lasminingrum, Melati Sudiro and Yussy Afriani Dewi

ORLHNS Departement, Padjadjaran University, Bandung, Indonesia

Corresponding author
Gracia Cintia Massie, massiegracia@gmail.com

## ABSTRACT

**Introduction**. Laryngeal carcinoma is a type of cancer that occurs in the laryngeal tissue with a high mortality rate. Fiber optic laryngoscopy examination with i-scan is a non-invasive technology used to visualize changes in the structure of the mucosa and blood vessels, aiding in better identification of malignancies. Despite its promising potential, the routine use of i-scan technology in examinations is not yet established in Indonesia, especially West Java.

**Objective**. To compare endoscopic findings using i-scan with histopathological results in patients with laryngeal carcinoma.

**Methods**. This study is an analytical prospective observational research with a cross-sectional design, followed by a concordance test analysis using accuracy tests and the kappa index. The data collected include information from all laryngeal tumor patients at the ORLHNS outpatient clinic at Dr. Hasan Sadikin General Hospital from July to December 2023.

**Results**. This prospective observational study evaluated i-scan endoscopy's diagnostic accuracy versus histopathology in 29 laryngeal carcinoma patients, demonstrating 96% sensitivity (95% CI [80.4–99.3%]) and 100% specificity (95% CI [39.8–100%]) with substantial histopathological agreement ($\kappa = 0.86$, 95% CI [0.61–1.00]). The technology outperformed white light endoscopy ($\kappa = 0.608$) in detecting malignancies, correctly identifying vascular patterns in 24/25 malignant cases (eight poorly differentiated, six moderately differentiated, five well-differentiated squamous cell carcinomas) and all four benign lesions. One false-negative involved a well-differentiated carcinoma, potentially due to obscuring edema or tumor positioning. These findings underscore i-scan's utility for precise malignancy detection and biopsy guidance in laryngeal carcinoma evaluation.

**Conclusion**. Our findings demonstrate concordance between i-scan endoscopic examination and histopathological results in laryngeal carcinoma evaluation. While this study provides preliminary evidence supporting the potential utility of i-scan technology for early lesion identification and biopsy targeting, further validation through larger-scale multicenter studies is warranted to confirm its clinical applicability.

## INTRODUCTION

Laryngeal carcinoma, characterized by the formation of malignant cells in the laryngeal tissues, ranks as the second most common head and neck malignancy globally, with a significant mortality rate. Global epidemiological data reveals varying presentations, with many cases originating in the supraglottis and presenting with non-specific symptoms such as throat pain and dysphagia, leading to delayed diagnosis and detection at advanced stages. Studies report that a significant proportion of laryngeal carcinoma cases (60.2%) are diagnosed at advanced stages (T3–T4), with notable lymph node metastasis rates, particularly in supraglottic cancer (28.8%). Major risk factors, including smoking (86.9% of patients) and alcohol consumption (43.2%), substantially contribute to disease progression and poor outcomes (*Koirala, 2015*; *Markou et al., 2013*; *Huang et al., 2024*).

According to GLOBOCAN 2020, laryngeal cancer ranks 20th in global incidence (ASR = 2.0 per 100,000) and 18th in mortality (ASR = 1.0 per 100,000), with significant geographical disparities. Europe demonstrates the highest burden, particularly in Central and Eastern Europe (ASR incidence = 3.6; mortality = 1.9) and Southern Europe (ASR incidence = 2.9), followed by North America and the Caribbean (ASR incidence = 4.0; mortality = 2.1). While Southeast Asia shows lower overall incidence, regions like South-Central Asia exhibit elevated mortality rates (ASR mortality = 1.7), reflecting potential disparities in early detection and treatment access (*Nocini et al., 2020*; *Farhat, Asnir & Yudhistira, 2023*; *Bray et al., 2018*; *Uddin et al., 2020*).

In laryngeal carcinoma, prolonged exposure to carcinogens induces chronic mucosal irritation, leading to genetic and epigenetic alterations in laryngeal epithelial cells. These changes manifest progressively as hyperplasia, dysplasia, carcinoma *in situ* (CIS), and invasive carcinoma. Dysplastic changes are classified histologically as low-grade or high-grade, with the latter carrying a higher risk of malignant transformation. The Ljubljana/WHO 2017 classification recognizes high-grade dysplasia and CIS as significant precursor lesions, with CIS presenting a 15–18% risk of progressing to invasive squamous cell carcinoma. Microinvasive carcinoma occurs when malignant cells breach the basement membrane, while invasive carcinoma involves stromal infiltration and potential metastasis (*Farhat, Asnir & Yudhistira, 2023*; *Cahyadi et al., 2016*; *Massie, Permana & Boesoirie, 2023*; *Bahadur & Malik, 2019*; *Chauhan et al., 2018*).

Angiogenesis plays a crucial role in malignancy, facilitating oxygen and nutrient supply essential for solid neoplasm survival. Through mechanisms including vasculogenesis, sprouting angiogenesis, intussusception, and vasculogenic mimicry, pathological vascular changes occur in the larynx, beginning with local blood vessel ectasia. These vessels become increasingly unstable and permeable, developing tortuous paths resembling varicose veins that serve as important diagnostic indicators (*Hawkshaw, Sataloff & Sataloff, 2014*; *Lee, Seong & Yong, 2021*).

While histopathological examination through microlaryngoscopic biopsy remains the gold standard for diagnosis, non-invasive technologies continue to evolve. Fiber optic laryngoscopy with I-scan technology represents a minimally invasive method for detecting malignant laryngeal lesions. This dynamic digital image processor combines high-powered

endoscopy with advanced image processing techniques, enhancing mucosal surface contrast, vascular patterns, and subtle abnormalities without compromising light intensity. The technology's three modes—surface enhancement (SE), contrast enhancement (CE), and tone enhancement (TE)—highlight different aspects of mucosal structure and vascular architecture, enabling more precise identification of premalignant and neoplastic changes characterized by distinctive microvascular patterns (*Nocini et al., 2020*; *Lee, Eun & Park, 2018*; *Miranda-Galvis et al., 2021*; *Bradford et al., 2020*; *Putri, Dewi & Dewayani, 2018*).

Laryngeal lesions observed through image-enhanced endoscopy can be categorized into seven distinct intraepithelial papillary capillary loop (IPCL) morphological patterns correlating with pathological progression. This classification system distinguishes between benign (Types I–III), precancerous (Type IV), and malignant (Va-Vc) categories based on vascular architecture changes, reflecting the progressive disruption of vascular integrity during carcinogenesis. The visualization of these microvascular transformations enables precise differentiation between various lesion types, directly informing biopsy targeting and staging accuracy (*Van Lierde et al., 2020*; *Nishimura et al., 2014*).

Multiple international studies have demonstrated I-scan's diagnostic utility in various malignancies, with research by *Lee, Seong & Yong (2021)* indicating higher sensitivity and accuracy compared to conventional white light in predicting neoplastic lesions. Nishimura's study further supports I-scan's considerable sensitivity and specificity in predicting small adenomatous polyps and malignancies (*Lee, Eun & Park, 2018*). Despite these promising results, widespread clinical application of I-scan technology for laryngeal carcinoma evaluation remains limited globally, representing a significant gap in implementation. Therefore, further research is needed to validate the compatibility of I-scan endoscopic examination with histopathological results in patients with laryngeal malignancies across diverse clinical settings worldwide.

## MATERIALS & METHODS

### Study population & design

The study included patients suspected of laryngeal carcinoma at Dr. Hasan Sadikin General Hospital. Inclusion criteria were suspected laryngeal malignancy cases. Patients were excluded if they had recurrent or residual laryngeal carcinoma, uncontrolled coagulopathy (INR >1.5), severe cardiopulmonary compromise precluding endoscopic evaluation, known contrast allergies affecting i-scan utilization, or inability to provide informed consent due to cognitive impairment. Additionally, patients with concurrent head/neck malignancies or those who had undergone prior laryngeal radiation/chemotherapy within six months were excluded to minimize confounding factors. The target population was Indonesian patients with suspected laryngeal malignancy, and the accessible population was those who come for treatment at the Hasan Sadikin General hospital's ORLHNS. Written informed consent was obtained from all participants prior to their inclusion in the study.

The study consecutively enrolled eligible patients presenting with suspected laryngeal malignancy to Dr. Hasan Sadikin General Hospital's ORLHNS clinic between July-December 2023 using non-probability sampling. Inclusion required symptomatic

presentation (laryngeal mass, dysphonia, dysphagia), while exclusion criteria excluded recurrent/residual carcinoma, coagulopathy (INR > 1.5), severe cardiopulmonary compromise, contrast allergies, cognitive impairment, recent laryngeal therapy, or concurrent head/neck malignancies. Three board-certified otolaryngologists ($\geq$5 years' endoscopic experience) independently assessed eligibility using Ni classification for i-scan interpretations, with final enrollment requiring consensus. This consecutive approach ensured clinical homogeneity through strict protocol adherence without active matching, maintaining temporal comparability across participants.

## Data collection and statistical analyses

Patients received an explanation of the aims and objectives of the research, along with details about the examination stages and techniques involved. Those who consented to participate in the study underwent anamnesis, physical examination, and fiber optic laryngoscopy with both white light and i-scan. The assessment and documentation of mucosal and vascular changes in the larynx were conducted and classified according to the Ni endoscopic classification. All subjects with suspected laryngeal malignancy underwent laryngeal biopsy *via* microlaryngoscopic approach. Biopsy samples were collected at sites exhibiting mucosal changes identified through i-scan.

All specimens will be fixed in formalin, embedded in paraffin, and stained with hematoxylin-eosin. Histopathological examination will be performed by two independent senior pathologists with expertise in head and neck malignancies, following WHO classification criteria for laryngeal squamous cell carcinoma. Inter-observer agreement will be calculated using Cohen's kappa coefficient. Discrepancies in diagnosis will be resolved through consensus review.

For i-scan image interpretation, three experienced otolaryngologists independently evaluated all endoscopic recordings. Each observer assessed the mucosal and vascular patterns according to the Ni classification system. To ensure standardization, observers underwent training sessions prior to the study using reference i-scan images. For cases where there was disagreement in interpretation, a consensus meeting was held where all three observers reviewed the images together to reach a final classification. Inter-observer agreement for i-scan interpretation was assessed using Fleiss' kappa coefficient to measure reliability across multiple raters. The final i-scan classification used for analysis was based on either unanimous agreement or the consensus decision reached after discussion

The i-scan examinations were performed using a PENTAX Medical videoendoscope system (EPK-3000 processor) with standardized SE + TE mode activation at 1.5−2.0 cm working distance, following a structured evaluation protocol by three blinded otolaryngologists applying the Ni classification system for vascular pattern analysis. Histopathological processing involved obtaining 3–5 mm biopsy specimens from i-scan-identified suspicious areas, fixed in 10% neutral buffered formalin, paraffin-embedded, sectioned at four $\mu$m, and stained with hematoxylin-eosin plus periodic acid-Schiff counterstain. Two senior pathologists independently assessed specimens using WHO diagnostic criteria, with Cohen's kappa coefficient calculation for interobserver agreement and multi-head microscope review sessions to resolve discrepancies.

The variables assessed in this study include age, gender, main complaint, tumor location, results of fiber optic laryngoscopy with white light, results of fiber optic laryngoscopy with i-scan, and results of histopathological examination.

Statistical analysis for categorical data was performed using the Chi-square test when the expected values were greater than or equal to 5 in at least 80% of the table cells. When this condition was not met, Fisher's Exact test was applied for 2 × 2 tables, and the Kolmogorov–Smirnov test was used for tables other than 2 × 2.

The chosen sample size aligns with similar studies in the field, further supporting its adequacy for achieving the study objectives (*Sakthivel et al., 2018*).

Diagnostic accuracy was assessed through calculations of sensitivity and specificity, each reported with corresponding 95% confidence intervals, alongside Cohen's kappa coefficient. Sensitivity represents the proportion of true positive malignant cases correctly identified by i-scan, while specificity indicates the proportion of true negative benign cases accurately classified. These metrics were derived from cross-tabulation of i-scan findings against histopathology results.

This research has been approved by the Medical Research Ethics Committee of Dr. Hasan Sadikin General Hospital, Bandung, West Java with ethical number LB.02.01/X.6.5/489/2023.

# RESULT

## Cohort characteristics

This research was conducted from July 2023 to January 2024. During this period, 31 patients with laryngeal tumors underwent endoscopic examination with i-scan, with two patients passing away before micro-laryngeal biopsy. Twenty-nine patients met the inclusion and exclusion criteria and underwent endoscopic examination using i-scan and histopathological examination at Dr. Hasan Sadikin General Hospital in Bandung. The mean age of the patients was 52.3 ± 17.8 years, with the most common age group being 56–65 years, comprising 26 male patients (89.7%) and three female patients (10.3%). The most common complaints were shortness of breath in 20 patients (69.0%) and hoarseness in seven patients (24.1%) (Table 1).

## Comparative analysis of diagnostic modalities

Out of the 29 patients who underwent fiber optic endoscopic examination using white light and i-scan, classified according to the Ni classification, the most common findings under white light were Type V, indicating malignancy, in 19 patients (65.5%). Similarly, under i-scan, Type V findings were observed in 24 patients (82.8%). Non-malignant findings accounted for 34.5% under white light and 17.2% under i-scan (Table 2).

## Diagnostic performance of i-scan technology

The diagnostic accuracy of i-scan technology demonstrated 96.0% sensitivity (24/25 true positives, one false negative) and 100.0% specificity (4/4 true negatives), confirming its high reliability in distinguishing malignant from benign laryngeal lesions.

**Table 1 Research subject characteristics.**

| Characteristic | Total N = 29 | Percentage (%) |
|---|---|---|
| **Age** | Esx | |
| <12 years | 2 | 6.9 |
| 12–16 years | 0 | 0 |
| 17–25 years | 1 | 3.45 |
| 26–35 years | 1 | 3.45 |
| 36–45 years | 3 | 10.34 |
| 46–55 years | 6 | 20.68 |
| 56–65 years | 8 | 27.59 |
| >65 years | 7 | 24.14 |
| Mean ± Std | 52.28 ± 17.816 | |
| Median | 56.00 | |
| Range | 5.00–77.00 | |
| **Gender** | | |
| Male | 26 | 89.7 |
| Female | 3 | 10.3 |
| **Chief complain** | | |
| Hoarse Voice | 7 | 24.14 |
| Shortness of Breath | 20 | 68.96 |
| Lump in the Neck | 1 | 3.44 |
| Dysphagia | 1 | 3.44 |
| **Tumor location** | | |
| Subglotis | 0 | 0 |
| Glotis | 7 | 24.14 |
| Supraglotis | 8 | 27.59 |
| Transglotis | 14 | 48.27 |

**Notes.**
Mean ± Std, Mean ± Standard Deviation; *N*, Number of subjects.

## Histopathological correlation of i-scan findings

In this study, non-malignant findings such as vocal cord polyps on i-scan endoscopy in two patients exhibited similar characteristics: thin and regular blood vessels, appearing slanted, and interconnected dendritic blood vessels visible, consistent with Type 1 (Fig. 1A). Another non-malignant finding, laryngeal papilloma, seen on i-scan endoscopy appeared as clusters of circles with central dots (Fig. 1B). Type IV findings, consisting of small brown dots, were observed in two patients with histopathological results of laryngeal papilloma. Malignant laryngeal features or Type V appeared as irregular blood vessels with larger brown dots or irregular spots resembling commas or worms in most patients in this study (Figs. 1C–1E). No Type III findings were observed. One i-scan endoscopy image showed slanted blood vessels and visible congested and widened dendritic blood vessels, consistent with Type II, but histopathology revealed malignancy as well-differentiated keratinizing squamous cell carcinoma, possibly influenced by factors like secretions, obscured tumor location due to surrounding tissue edema, or tumor pushing, or well-differentiated histopathology.

**Table 2  Endoscopic examination results based on Ni classification.**

| Characteristic | Total<br>N = 29 | Percentage<br>% |
|---|---|---|
| **White light** | | |
| Type I | 2 | 6.9 |
| Type II | 3 | 10.34 |
| Type III | 1 | 3.45 |
| Type IV | 4 | 13.79 |
| Type V | 19 | 65.52 |
| **I-Scan** | | |
| Type I | 2 | 6.9 |
| Type II | 1 | 3.45 |
| Type III | 0 | 0 |
| Type IV | 2 | 6.9 |
| Type V | 24 | 82.76 |

**Notes.**

N, Total number of cases/samples examined.
Type I–V refers to Ni Classification system for endoscopic mucosal patterns.

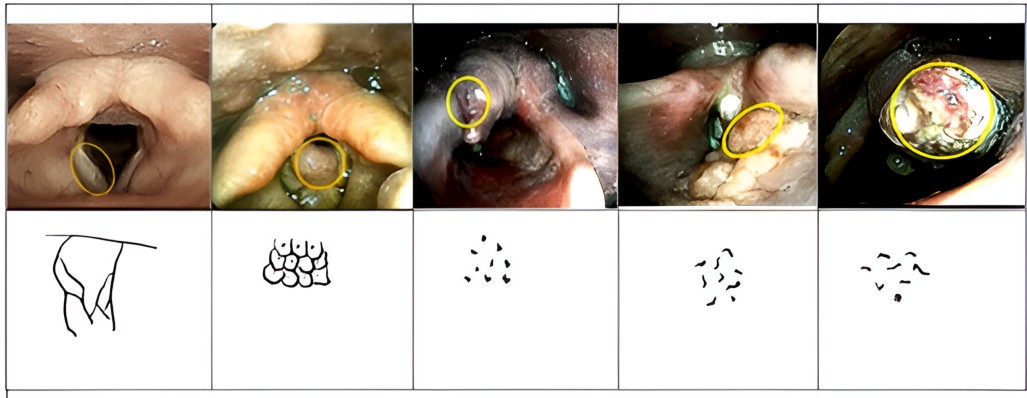

**Figure 1  Overview of endoscopic examination results using i-scan.** (A) Vocal fold nodule; (B) laryngeal papilloma C1-3. Malignant neoplasm on endoscopic examination using i-scan.

## Histopathological examination results

Of the 29 patients who underwent histopathological examination *via* biopsy using microlaryngoscopy, malignant neoplasms were detected in 25 patients (86.2%). These comprised poorly differentiated conventional squamous cell carcinoma (SCC) in 24.1% of cases, moderately differentiated conventional SCC in 20.7%, and well-differentiated SCC in 17.2%. Additionally, benign neoplasms were found in four patients (13.8%), including laryngeal papilloma in 6.9% of cases and vocal cord polyps in 6.9% (Table 3).

## Compatibility between white light endoscopy and histopathology

Out of the 19 patients whose endoscopy results using white light depicted malignancy, all also had histopathology results indicating malignant neoplasms. Among the 10 patients

**Table 3  Histopathology results.**

| Histopathology result | Total N = 29 | Percentage (%) |
|---|---|---|
| **Malignant neoplasm** | | |
| – SCC poorly differentiated | 7 | 24.14 |
| – SCC moderately differentiated | 6 | 20.69 |
| – SCC well differentiated | 5 | 17.24 |
| – Keratinizing SCC | 1 | 3.45 |
| – Keratinizing SCC well differentiated | 1 | 3.45 |
| – Keratinizing SCC poorly differentiated | 1 | 3.45 |
| – Non Keratinizing SCC poorly differentiated | 1 | 3.45 |
| – Non Keratinizing SCC moderately differentiated | 1 | 3.45 |
| – Papillary SCC | | |
| – Basaloid SCC | | |
| **Benign neoplasm** | | |
| – LarynxPapilloma | 2 | 6.9 |
| – Vocal cord polyp | 2 | 6.9 |

Notes.
SCC, Squamous Cell Carcinoma; N, Number of cases.

**Table 4  Sutaibility between endoscopy results using white light and histopathology results.**

| Variable | Histopathology result | | Kappa value |
|---|---|---|---|
| | **Malignant N = 25** | **Benign N = 4** | |
| **Hasil Endoskopi dengan White light** | | | |
| Malignant | 19 (76.0%) | 0 (0.0%) | 0.608 (95% CI [0.35–0.86]) |
| Benign | 10 (24.0%) | 4 (100.0%) | |

Notes.
N, Number of samples; CI, Confidence Interval.

whose endoscopy results using white light showed non-malignancy, histopathology revealed malignant neoplasms. Additionally, in four patients whose white light endoscopic examination results did not indicate malignancy, histopathological results showed benign neoplasms.

The statistical analysis demonstrated significant diagnostic concordance between endoscopic modalities and histopathology. White light endoscopy showed moderate agreement with histopathological findings, evidenced by a Cohen's kappa coefficient of 0.608 (95% CI [0.37–0.85]). The precision of these measures is reflected in their respective confidence intervals, with i-scan's narrower interval (despite upper bound truncation at 1.00) suggesting more robust reliability than white light examination (Table 4).

## Compatibility between i-scan endoscopy and histopathology

In 24 patients whose endoscopic examination using i-scan depicted malignancy, their histopathology results also confirmed malignant neoplasms.

**Table 5  Compatibility between endoscopy results using I-Scan and histopathology results.**

| Variable | Histopatology result | | Kappa value |
| --- | --- | --- | --- |
| | Malignant N = 25 | Benign N = 4 | |
| Endoscopy result with I-Scan | | | 0.86  (95% CI [0.61–1.00]) |
| Malignant | 24 (96.0%) | (0.0%) | |
| Benign | 1 (4.0%) | 4 (100.0%) | |

Notes.
  N, Number of samples;  CI, Confidence Interval.

There was one patient whose endoscopy results using i-scan indicated non-malignancy, but histopathological analysis revealed a malignant neoplasm, namely well-differentiated keratinizing squamous cell carcinoma. Additionally, in four patients whose endoscopic examination results using i-scan were non-malignant, their histopathology results also showed benign neoplasms. I-scan technology exhibited substantial agreement with histopathology, achieving a kappa coefficient of 0.86 (95% CI [0.61–1.00]) (Table 5).

## DISCUSSION

A total of 29 patients underwent endoscopic i-scan and histopathological examination at Dr. Hasan Sadikin General Hospital. The mean age was 52.3 ± 17.8 years, with the majority aged 56–65; 89.7% were male and 10.3% female, consistent with previous findings that laryngeal malignancy predominantly affects older males (*Nocini et al., 2020*; *Farhat, Asnir & Yudhistira, 2023*; *Bray et al., 2018*; *Uddin et al., 2020*; *Putri, Dewi & Dewayani, 2018*; *Koroulakis & Agarwal, 2023*). Internationally, the male-to-female ratio ranges from 30:1 to 5:1, though increasing rates in women have been observed, likely due to rising tobacco and alcohol use and possible hormonal influences (*Miranda-Galvis et al., 2021*; *Bradford et al., 2020*; *Putri, Dewi & Dewayani, 2018*). Similar demographic patterns are seen globally and in prior studies at this hospital (*Nocini et al., 2020*; *Farhat, Asnir & Yudhistira, 2023*; *Bray et al., 2018*; *Uddin et al., 2020*; *Putri, Dewi & Dewayani, 2018*; *Koroulakis & Agarwal, 2023*).

Most patients presented with shortness of breath (68.96%) and hoarseness (24.14%), consistent with studies in Thailand and at Dr. Hasan Sadikin General Hospital Bandung that found these symptoms common, particularly in advanced-stage cases. Tumor location was mainly transglottic (48.27%), which contrasts with findings by *Nocini et al. (2020)* and US data, where most laryngeal malignancies are found in the supraglottic or glottic areas (*Aliye & Nour, 2023*).

I-scan is an advanced endoscopic image-enhancement technology from PENTAX, Japan, that facilitates detailed vascular assessment without specialized magnifying scopes. Using the Ni classification of IPCL morphology, types I–IV typically indicate benign or preneoplastic lesions, while type V is associated with malignancy. Among 29 patients, type V was most common on both white light (65.52%) and i-scan (82.76%) endoscopy, with non-malignant (types I–IV) seen in 34.48% (WL) and 17.24% (i-scan). These results are consistent with *Ahmadzada et al. (2020)* who found a majority of type V lesions represented

malignancy, and with studies noting "brown dots" reflect neoplastic angiogenesis (*Mehlum et al., 2018*; *Arthur et al., 2023*). Other research, such as Lee et al., demonstrates i-scan's capacity to distinguish vascular patterns correlating with dysplasia severity. Overall, i-scan enhances detection and characterization of laryngeal lesions compared to conventional techniques (*Mehlum et al., 2018*; *Arthur et al., 2023*; *Ahmadzada et al., 2020*).

Among the 29 patients who underwent microlaryngoscopic biopsy, 25 (86.2%) had malignant neoplasms—all conventional SCC (poorly differentiated 24.14%, moderately differentiated 20.69%, well differentiated 17.24%)—and 4 (13.8%) had benign lesions (laryngeal papilloma 6.9%, vocal cord polyps 6.9%). This matches global data that most laryngeal malignancies are squamous cell carcinomas. Notably, one patient showed a non-malignant i-scan (type II) but was diagnosed as well-differentiated keratinizing SCC on histopathology. Tumor differentiation impacts prognosis, with poorly differentiated SCC linked to higher recurrence and mortality rates, as also shown by Zhang et al. Further research is needed on the influence of histopathological type and differentiation on i-scan results (*Nocini et al., 2020*; *Bahadur & Malik, 2019*; *Zhang et al., 2021*).

The conformity test showed moderate agreement between white light endoscopy and histopathology (kappa = 0.608), as a kappa between 0.40–0.74 reflects moderate suitability. While white light is widely used to assess premalignant laryngeal lesions, its diagnostic accuracy can be limited, especially when mucosal changes such as leukoplakia obscure visualization. Adjunct technologies like i-scan can help detect lesions not clearly seen with white light alone, and diagnostic accuracy ultimately depends on both the clinician's expertise and the technology used (*Bahadur & Malik, 2019*; *Koroulakis & Agarwal, 2023*).

Thick white mucosal patches hinder malignancy detection by obscuring vascular features. In our study, white light endoscopy showed only moderate agreement with histopathology ($\kappa = 0.608$), while i-scan demonstrated substantial agreement ($\kappa = 0.86$), despite one false-negative i-scan result in a well-differentiated SCC hidden by edema. Keratinized surfaces can reduce mucosal reflectance by 62%, limiting lesion visibility. Similar challenges and imaging advances have been noted in gastrointestinal and dermatological studies, where enhanced techniques like i-scan and dual imaging modes improved neoplasia detection . Our results support routine use of i-scan's SE/TE modes to optimize visualization, especially in high-risk leukoplakia, to address the 22% prevalence of occult malignancy (*Uddin et al., 2020*; *Chauhan et al., 2018*).

I-scan technology enhances diagnostic accuracy in aerodigestive endoscopy by improving detection of mucosal and vascular changes. In this study, i-scan predicted malignancy in 24 patients, all confirmed by histopathology, with only one false-negative case (well-differentiated keratinizing SCC). Four non-malignant i-scan cases were also confirmed benign. The kappa value for agreement between i-scan and histopathology was 0.86, indicating high concordance, which is superior to white light endoscopy. I-scan showed 96% sensitivity and 100% specificity for malignancy, matching results from previous gastrointestinal studies. This improvement is attributed to i-scan's advanced imaging modes (SE, CE, TE), which enhance surface and vascular details without additional dyes. Other studies have also demonstrated i-scan's high accuracy for detecting laryngeal malignancy and for guiding biopsies, reinforcing its value as a diagnostic tool (*Lee, Eun &*

*Park, 2018*; *Van Lierde et al., 2020*; *Nishimura et al., 2014*; *Sakthivel et al., 2018*; *Koroulakis & Agarwal, 2023*; *Iacucci et al., 2016*).

The study's limitations extend beyond sample size constraints to encompass methodological and contextual factors that warrant careful consideration. While the small cohort ($n = 29$) limited histological subgroup analyses, the single-center design at a tertiary referral hospital may restrict generalizability to broader populations due to potential selection bias in patient demographics and disease severity profiles. Although practitioners had ≥5 years' endoscopic experience, interobserver variability in i-scan interpretation—despite consensus protocols—could influence diagnostic consistency, particularly given the technology's novelty in this clinical setting. Furthermore, the exclusion of patients with recurrent/residual disease or prior therapies creates a "pure" cohort that may not fully represent real-world clinical complexity. These factors collectively underscore the need for cautious interpretation of the reported high concordance ($\kappa = 0.86$), as institutional expertise in both i-scan utilization and histopathological assessment might artificially enhance performance metrics compared to routine practice environments. Future multi-center validation should address these limitations through standardized training protocols and inclusion of more heterogeneous patient populations.

## CONCLUSION

Our findings demonstrate substantial diagnostic concordance between i-scan endoscopic evaluation and histopathological examination in laryngeal carcinoma detection, evidenced by $\kappa$ agreement of 0.86 (95% CI [0.61–1.00]) and 100% specificity for vascular pattern characterization. The technology's capability to identify vascular abnormalities in 24/25 malignant cases supports its potential as an adjunctive tool for lesion characterization. However, several methodological constraints warrant consideration, including the single-center West Java cohort ($n = 29$), limited practitioner cohort (three operators), and challenges in interpreting subtle mucosal changes—as exemplified by one false-negative case of well-differentiated SCC obscured by edema—necessitating cautious interpretation of these results. While our observations of vascular pattern recognition parallel prior applications in gastrointestinal endoscopy, the absence of multi-institutional validation and standardized training protocols limits the generalizability of our findings. These data position i-scan as a complementary visualization tool rather than a replacement for histopathological examination, with broader clinical integration contingent upon confirmation through large-scale multicenter trials encompassing diverse populations and histopathological subtypes. Future research should aim to establish diagnostic consistency across varying operator experience levels through optimized training programs, with particular attention to improving detection of edema-obscured malignancies, which affected 3.4% of assessments in our study.

### Funding

The authors received no funding for this work.

## Competing Interests

The authors declare there are no competing interests

## Author Contributions

- Gracia Cintia Massie conceived and designed the experiments, performed the experiments, analyzed the data, prepared figures and/or tables, authored or reviewed drafts of the article, and approved the final draft.
- Agung Dinasti Permana conceived and designed the experiments, performed the experiments, analyzed the data, prepared figures and/or tables, authored or reviewed drafts of the article, and approved the final draft.
- Shinta Fitri Boesoirie conceived and designed the experiments, analyzed the data, prepared figures and/or tables, authored or reviewed drafts of the article, and approved the final draft.
- Lina Lasminingrum analyzed the data, authored or reviewed drafts of the article, and approved the final draft.
- Melati Sudiro analyzed the data, authored or reviewed drafts of the article, and approved the final draft.
- Yussy Afriani Dewi analyzed the data, authored or reviewed drafts of the article, and approved the final draft.

## Human Ethics

The following information was supplied relating to ethical approvals (*i.e.*, approving body and any reference numbers):

The Hasan Sadikin General Hospital granted Ethical approval to carry out the study within its facilities.

## Data Availability

The raw measurements are available in the Supplementary Files S1 and S2.

## Supplemental Information

Supplemental information for this article can be found online at http://dx.doi.org/10.7717/peerj.19552#supplemental-information.

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
