# Peer review of "Compatibility of endoscopic examination using i-scan technology with histopathology results in laryngeal carcinoma: prospective observational study"

_PeerJ, doi:10.7717/peerj.19552_

## Round 0.1 · original submission · Major Revisions

Please address the detailed comments from both reviewers.

Reviewer 1 ·

Basic reporting

The manuscript entitled Compatibility of endoscopic examination using I-Scan technology with histopathology results in laryngeal carcinoma investigating the use of I-Scan technology in the diagnosis of laryngeal carcinoma. The study compared the findings of endoscopic examinations using I-Scan with histopathology results in a group of patients with laryngeal tumors. The results showed a high level of concordance between the two methods, suggesting that I-Scan can be a valuable tool for early detection and guiding biopsy procedures in laryngeal cancer. While the study may still provide valuable insights, it's crucial to consider the following factors when evaluating its findings:


1. The absence of histopathology images in a study comparing I-Scan technology to histopathology results in laryngeal carcinoma is a significant limitation. The images should be included, How was the histopathological reference standard established? Were there any inter-observer variations in the histopathological diagnosis?

2. How does I-Scan compare to other advanced endoscopic techniques (e.g., narrow-band imaging, autofluorescence) in detecting laryngeal carcinoma?

3. Can I-Scan be used to predict the prognosis of laryngeal cancer based on specific microvascular patterns?

4.How does I-Scan performance vary among different histological subtypes of laryngeal cancer?

5. endoscopic findings using i-scan and histopathological results had a kappa value of 0.608. While the kappa value of 0.608 indicates moderate agreement, is this sufficient for clinical decision-making, especially in a disease as serious as laryngeal carcinoma?

6. Were sensitivity and specificity calculated to assess the diagnostic accuracy of i-scan?

Experimental design

Is the sample size of 29 patients sufficient to draw robust conclusions about the concordance between i-scan and histopathology?

Validity of the findings

How was inter-observer variability addressed in the interpretation of both i-scan images and histopathology slides? Were multiple observers involved, and if so, was there a consensus-based approach or a statistical method to assess the level of agreement?

Reviewer 2 ·

Basic reporting

Title & Abstract
Title: Whilst the title provides the clinical components for the paper, there is a need to clearly articulate the study design. I recommend the authors revise the title to something such as “Compatibility of Endoscopic Examination Using I-Scan Technology with Histopathology Results in Laryngeal Carcinoma: prospective observational study”.
Abstract:
Introduction: PDF Page 5, Lines 14-23: This section is very long and detracts from the flow of the Abstract. There is opportunity to present more information on the Methods within the Abstract word count. I therefore recommend the authors consider whether sentences such as Line 15-19 be removed as this level of epidemiological context is not required in the Abstract.
PDF Page 5, Line 24: I recommend ‘Methods’ is presented in BOLD as per the other section headings. This will improve readability.
PDF Page 5, Line 28: I recommend incorporating further information on the i-scan technology and type of histopathology results should be included. I also recommend adding in one sentence on statistical analysis approach. The word count gained through removing redundant epidemiological information in the Introduction section of the Abstract, can be used more effectively here.
Results: PDF Page 5, Line 32-33: I recommend the authors provide 95% confidence interval data for both the concordance between the i-scan and histopathological results.
Results: PDF Page 5, Line 33: I recommend the authors revise the sentence “The kappa value for i-scan showed high concordance with a value of 0.86” – it is not clear what this concordance is i.e. with histopathological results as reported in the previous sentence or some form of reliability data. I recommend this is explicitly stated.
Results: PDF Page 5, Line 33: I recommend the authors provide the 95% confidence intervals for the kappa 0.86 value. This would further improve clarification of reporting.
Conclusions, PDF Page 5, Line 35-37: The conclusions are currently too strong based on the small number of cases examined in this study. The results need to be more cautiously presented in the conclusion, with emphasis that this study offers indicative findings or exploratory findings to justify further, larger, examination. I therefore recommend the authors revised the final sentence of the Abstract Conclusions section and ‘tone down’ the clinical utility until more robust data is presented in further study.

Introduction
PDF Page 6, Line 63-65 AND Line 65-67 AND Line 71-77: I recommend the authors provide a citation to support this statement on prevalence of laryngeal carcinoma and disease basis. Papers such as the following may be helpful to support these statements:
• Galli J, Cammarota G, Volante M, De Corso E, Almadori G, Paludetti G. Laryngeal carcinoma and laryngo-pharyngeal reflux disease. Acta Otorhinolaryngol Ital. 2006 Oct;26(5):260-3

• Koirala K. Epidemiological Study of Laryngeal Carcinoma in Western Nepal. Asian Pac J Cancer Prev. 2015;16(15):6541-4
• Markou K, Christoforidou A, Karasmanis I, Tsiropoulos G, Triaridis S, Constantinidis I, Vital V, Nikolaou A. Laryngeal cancer: epidemiological data from Νorthern Greece and review of the literature. Hippokratia. 2013 Oct;17(4):313-8.
• Huang J, Chan SC, Ko S, Lok V, Zhang L, Lin X, Lucero-Prisno DE 3rd, Xu W, Zheng ZJ, Elcarte E, Withers M, Wong MC. Updated disease distributions, risk factors, and trends of laryngeal cancer: a global analysis of cancer registries. Int J Surg. 2024 Feb 1;110(2):810-819
• Odell E, Eckel HE, Simo R, Quer M, Paleri V, Klussmann JP, Remacle M, Sjögren E, Piazza C. European Laryngological Society position paper on laryngeal dysplasia Part I: aetiology and pathological classification. Eur Arch Otorhinolaryngol. 2021 Jun;278(6):1717-1722
PDF Page 9, Lines 111-129: The depth of information provided in the Introduction on the i-scan technology (Lines90-110) is helpful but the corresponding information on endoscopic examination and histology is somewhat overpowering and I think this could be abridged to offer a summary of the information, but to lesser depth. This could improve the flow and readability of the Introduction. I therefore recommend the authors review Lines 111-131.
Introduction, PDF Page 10, Line 140: Following on from acknowledging the limited clinical application of i-scan technology as a diagnostic tool for laryngeal cancers, I recommend the authors make an explicit statement on whether this has been reported in the literature or not. A clear statement such as this would improve the rationale for this paper.
Overall, the language of the manuscript is acceptable however there is a need to review the citation use with the comments provided in the review to promote use of evidence citations to better support the statements made.

Figures & Tables
There are no issues with respect to modification or image integrity of Tables or Figures. However, all tables are lacking footnotes to explain the abbreviations.
Table 1: Some of the terms are not presented in English in this table, principally Gender and Chief complaint.
Table 4 and 5: 95% confidence interval data should be presented with the Kappa value in these tables.
Please ensure that the tables are revised as per the comments provided. Please address it.

Experimental design

Material and Methods
PDF Page 10, Lines 150-151: Were there any other eligibility criteria? I recommend the authors consider whether there are any further contraindications to performing this assessment. This is important as the reporting of this technology in this context is limited, and clearly defining when and when not this is indicated, would be important. This would also aid the interpretation of external validity for this cohort. I recommend this clarification is made in the paper.
PDF Page 11, Lines 156-159: There is currently insufficient information provided in the paper on how the cohort was derived. There is reference to non-probability sampling and a suggestion of comparability. I recommend the authors explain in greater detail how the sample was gained, what the characteristics were for determining matching or comparability in participant presentation and who performed this. I recommend this is incorporated into this section of the Methods to improve clarification of reporting.
PDF Page 11, Line 162-169: I recommend the authors review the grammar in this sentence. I think this should be “Patients received an explanation….” rather than “Patients will receive an explanation…” – this continues in this paragraph i.e. “Those who consent to participate in the study will undergo” should be “…in the study underwent…” - I recommend this whole paragraph is reviewed for tense of writing.
PDF Page 12, Line 173: I recommend the removal of the sentence “The analysis conducted must be appropriate to the type of research problem and the data used” – this is redundant and not required.
PDF Page 12, Line 175: I recommend the authors state what the ‘conditions’ are for being met for the adoption of the chi-squared test. Do they mean data distribution or size of sample? I think this may be stated in Line 177 but this needs to be moved directly to Line 175 to ensure this is understood by the reader.
PDF Page 12, Line 177: There is no justification for the sample size. I recommend some statement to justify the number of participants is needed, preferably based on a sample size calculation to offer assurance that the data presented were sufficiently powerful to answer the research questions.
PDF Lage 12, Line 179: There is insufficient information provided on the actual i-scan procedure and histological results. This may be because a large amount of the information is presented in the Introduction (as acknowledged earlier). For reporting and understanding what happened to the study participants in this paper, I recommend the authors reconsider the ordering of the text and provide some text in the Methods section outlining what happened to the paper from a biopsy and histopathological examination perspective. This would improve the understanding of the methods and overall paper’s interpretation.

Validity of the findings

Results
PDF Page 12-13: The results are plausible and credible. I recommend the data in this section be presented through subheadings such as Cohort Characteristics, Comparative Analysis, Histopathological Results etc. etc. This will improve the reporting in Page 12 and 13, thereby improving the interpretation of the study findings.
PDF Page 12, Line 182: The data of the research was already presented in Line 150. I recommend only stating this once in the paper as a second time is redundant.
PDF Page 12, Line 186-187: I recommend presenting age data to one decimal place i.e. mean 52.3 (SD: 17.8). There is no need to present the frequency of the most common age group. This would aid clarity of presentation. Furthermore, the term ‘average age’ should be replaced with ‘mean age’ if this is appropriate.
PDF Page 12, Line 189-190: I recommend only presenting the common complaints to two decimal places. This will improve clarity of presentation.
PDF Page 12-13, Lines 192-196: The most common findings from i-scan assessment may be better presented through a histogram or pie-chart. I recommend visual reporting of this data to aid interpretation.
PDF Page 13, Lines 212-216: I recommend presenting data on histopathological examination to one decimal place only in the text as more is not required to communicate this data.
PDF Page 14, Laine 222-225: The data on compatibility between the tests is the principal analysis. This should be presented as 95% confidence interval data rather than range data for the Kappa value. I recommend these data to be presented as such to provide an indication on the precision of the results.
PDF Page 14, Line 231-233: The sentence “Statistical analysis showed a high level of compatibility between the Endoscopy examination using I-Scan and the Histopathology Results, with a kappa value of 0.86. This kappa value exceeding 0.75 indicates high compatibility” requires further data. For example, there is a need to offer the 95% confidence intervals around the kappa value 0.86. There is also a need to provide a citation to underpin the statement that exceeding kappa 0.75 indicates high compatibility.



Discussion
PDF Page 14, Lines 235-237: There is no need to specifically report the epidemiology of the patient groups again in this paragraph as this is already clearly presented in the Introduction. Similarly, there is no need to re-present the results of the cohort i.e. PDF Page 15, Lines 239-240 as this is already clearly communicated in the Results. I therefore recommend a clearer, more direct summary of the findings be presented for the principal analysis i.e. comparative analysis of i-scan results with histopathological findings.
PDF Page 15, Line 241: I recommend the authors add a citation number for Koroulakis et al's paper and Putri et al in Line 242. Presenting the reference numbers at the end of the paragraph alone is not sufficient and makes interpretation more challenging.
PDF Page 15, Lines 249-259 and PDF Page 16, Lines 260-284: This text relates to the disease process and epidemiology. The purpose of this study was to assess the comparability of the i-scan technology with histopathological examination. Therefore, the Discussion should be focused on interpreting the findings or reasons for comparability or not for this investigatory approach. These paragraphs are therefore redundant as do not interpret but provide further descriptive content. I recommend the authors reflect on the value of these paragraphs and focus the Discussion to more of an interpretation of the results and their clinical or research utility, rather than providing a literature review on the pathophysiology and epidemiology of laryngeal carcinoma.
PDF Page 19, Line 334-356: This paragraph is critical to the Discussion. It provides a valuable reflection to begin to explore how robust the data are and why there may have been differences in response to histopathological examination. The statement “Sometimes imaging can give false negative results because the underlying mucosa may be covered by a thick white patch, thus preventing diagnosis” is useful – I recommend the authors consider reasons for this and examples of other literature where visual field is important and white light results. This would strengthen this section.
PDF Page 20, Line 374: The Discussion lacks a limitations section. The authors should communicate the shortcomings in their study design. Most important are sample size, cohort representation, single center issues and practitioner experience. Examining the impact of these on the overall outcome is essential as currently there is insufficient reflection on how these issues may interact with the conclusions drawn. I recommend this paragraph be included.

Conclusion
PDF Page 21, Lines 376-379: The conclusions are too strong based on the study’s limitations. The impact of small sample size, single centre data collection and limited number of practitioners involved impacts on the power of the analyses and the generalizability or external validity of these findings. I would suggest that these data offer an indication on the comparability of i-scan to histopathological examination but further, larger-scale examination is warranted. I recommend this amendment both in this section and also reflected in the Abstract’s Conclusion section.

Additional comments

no comment

---

## Round 0.2 · Minor Revisions

The authors are requested to carefully revise the manuscript and answer the questions raised by the reviewers.

Reviewer 1 ·

Basic reporting

All queries resolved.

Experimental design

All queries resolved.

Validity of the findings

All queries resolved.

Additional comments

All queries resolved.

Reviewer 2 ·

Basic reporting

Title & Abstract
Title: This has been revised and is appropriate.
Abstract: Page 6, Lines 18-41: This has been revised and the previous points addressed

Introduction
Introduction: Page 9, Line 66 - Page 13, Line 177: The Introduction remains particularly long. The previous comments recommended making the Introduction more concise and abridged. This has not been fully executed. The authors still provide too much context in this revised paper. I strongly recommend the authors abridge the Introduction, aiming to provide the context and rationale across 1.5 sides of A4 in total. Currently, the depth is too much and this impacts on the readability of this paper.
Introduction, Page 13, Line 169-177: The justification for the study with reference to “particularly at Dr. Hasan Sadikin General Hospital” is inappropriate as it questions the generalizability of the study. I recommend the authors take this ‘local’ perspective out and present a far more global perspective on why this study is needed and what this study provides.
Introduction, Page 9, Line 66 - Page 13, Line 177: The issues on missing citations have been addressed. I have nothing further to recommend on this section.

Figures & Tables
The figures and tables are appropriate. There are no issues regarding legibility or modifications.

Experimental design

Material and Methods
The authors have addressed both Reviewer 1’s and my comments largely in this section. I have three remaining points listed below.
Methods, Page 16, Line 239-242: The authors state “Statistical analysis for categorical data will be tested using the Chi-square test when the expected values are greater than or equal to 5 in at least 80% of the table cells. When this condition is not met, the Exact Fisher test will be used for 2 x 2 tables, and the Kolmogorov-Smirnov test will be used for tables other than 2 x 2” – this is correct, but I recommend that grammatically should be presented in the third person passive tense.
Methods, Page 16, Line 243-249: The justification made for the same size is not appropriate. This is a generalization. This is not specific to this research question and does not provide any valuable justification for the author’s decision-making on what the planned or appropriate sample size was. The authors state “Additionally, the chosen sample size aligns with similar studies in the field, further supporting its adequacy for achieving the study objectives” – this statement would be sufficient to justify their sample size of they did not perform an a priori sample size, but I recommend the authors provide citations from previous studies in the field which they are referencing here, to support such a statement. The other text in this paragraph would then be redundant and could be deleted for clarity of reporting.
Results, Page 17, Line 250-254: In response to Reviewer 1, I welcome the additional data on sensitivity and specificity but recommend the authors refer to 95% confidence being presented for these data both in the Methods and in the Results. This would improve the robustness of reporting.

Validity of the findings

Results
The authors have largely addressed the points originally raised. There remain two points to address, as listed below.
Results, Page 17, Line 264 – Page 19, Line 300: The authors have not comprehensively addressed the points on decimal points. I recommend the authors review all standard deviation and percentage decimal points in this section and correct them to one decimal point to improve robustness of reporting.
Results: The interobserver reliability is not clearly presented. I recommend the authors provide these data, with 95% confidence intervals, within a subheading in the Results section.

Discussion
Discussion, Page 20-27: This is too long. The authors should consider abridging the text to present their interpretation across 3 to 4 sides of manuscript only. The current text remains difficult to navigate and could be more concise. I recommend this to improve the strength of reporting this paper.
Discussion, Page 20, Line 327: I believe the authors should correct “52.28 ± 17,816 years” to “52.3 ± 17.8” for accuracy of reporting and correct the comma to a full stop for 17,816 as I suggest this is a typo.
Discussion, Page 22, Line 373: The authors state “The results of this study are in line 374 with Ahmadzada et al's research were based on the Ni classification, 23.0% of the results were 375 77.0% of the patients were graded type V” but do not provide a citation number for Ahmadzada et al's. This should be corrected here, and I recommend all other references to studies should also be supplemented with a citation number in the Discussion to improve reporting. This includes, Page 23, Line 379-381, Line 381-838, Page 25 , Line 439-441, for example.
Discussion, Page 27, Line 471-484: The limitations section is improved. I have nothing further to recommend.
The references in the paper should be corrected. Page 9, Line 68 and 75 for example should be cited as numbers and therefore the reference list will need to be corrected accordingly as well.

Conclusion
Conclusions, Page 27, Line 487 to Page 28, Line 501: The revised conclusions are appropriate. I have nothing further to recommend.

Additional comments

None

---

## Round 0.3 · accepted · Accept

After revisions, all reviewers agreed to publish the manuscript. I believe the manuscript can be published once the author supplements the medical ethics approval number in the text.

Reviewer 2 ·

Basic reporting

Title & Abstract
Title
The title is clear and revised as recommended. I have nothing further to recommend.
Abstract
Abstract: Page 6 – This is clear and appropriately corrected. I have one minor point of clarification below.
Abstract: Methods, Page 6, Line 24: The sentence “This study is an analytical prospective observational research” should be revised to “This study is an analytical prospective observational research study…” as the current sentence is not sufficiently clear.

Introduction
Introduction: Page 9-11: The authors have abridged the Introduction without losing clarity. The rationale offers a more global perspective which is appropriate. The Introduction offers a clear background to the study with appropriate evidence underpinning the text. This is contemporary and accurately referenced. I have nothing further to recommend in relation to this section.

Figures & Tables
The tables and figures are clear and legible.

Experimental design

Material and Methods
Page 11-14: The authors have addressed the final points related to the statistical analysis reporting. There is now also clarity on how the sample size was determined, with a relevant citation provided. Overall, the Methods are now sufficiently clear and robustly presented. I have nothing further to recommend in this section.

Validity of the findings

Results
Page 15-17: The authors have provided the requested 95% confidence interval data. The interpretation of this is appropriate. Overall, the Results section is clear and answers the research question posed. I have nothing further to recommend.
Page 17: Line 251: The authors should correct “evidenced by a Cohen's kappa coefficient of 0.608” to “evidenced by a Cohen's kappa coefficient of 0.61” for clarity. All other points on decimal point reporting have been addressed from the previous review.

Discussion
Page 18-21: The authors have abridged the Discussion as recommended in the previous review without losing the meaning and impact of this section. The corrections and revisions highlighted in the last review have been addressed. The text is appropriately interpreted and grounded in sufficient relevant literature. I have nothing further to recommend.

Conclusion
Conclusions: Page 21-22: The revised conclusions remain appropriate. I have nothing further to recommend in relation to this section.